# Epigenetic Regulation of the Renin–Angiotensin–Aldosterone System in Hypertension

**DOI:** 10.3390/ijms25158099

**Published:** 2024-07-25

**Authors:** Yoshimichi Takeda, Masashi Demura, Takashi Yoneda, Yoshiyu Takeda

**Affiliations:** 1Endocrinology and Metabolism, Saiseikai Kanazawa Hospital, Kanazawa 920-0353, Japan; aldo_takeda@yahoo.co.jp; 2Department of Hygiene, Graduate School of Medical Science, Kanazawa University, Kanazawa 921-8641, Japan; m-demura@med.kanazawa-u.ac.jp; 3Institute of Liberal Arts and Science, Kanazawa University, Kanazawa 921-8641, Japan; endocrin@med.kanazawa-u.ac.jp; 4Department of Health Promotion of Medicine of the Future, Graduate School of Medical Science, Kanazawa University, Kanazawa 921-8641, Japan; 5Hypertension Center, Asanogawa General Hospital, Kanazawa 910-8621, Japan

**Keywords:** renin, angiotensin, aldosterone, angiotensin-converting enzyme, epigenetics, hypertension

## Abstract

Activation of the renin–angiotensin–aldosterone system (RAAS) plays an important pathophysiological role in hypertension. Increased mRNA levels of the *angiotensinogen angiotensin-converting enzyme*, angiotensin type 1 receptor gene, *Agtr1a*, and the aldosterone synthase gene, *CYP11B2,* have been reported in the heart, blood vessels, and kidneys in salt-sensitive hypertension. However, the mechanism of gene regulation in each component of the RAAS in cardiovascular and renal tissues is unclear. Epigenetic mechanisms, which are important for regulating gene expression, include DNA methylation, histone post-translational modifications, and microRNA (miRNA) regulation. A close association exists between low DNA methylation at CEBP-binding sites and increased *AGT* expression in visceral adipose tissue and the heart of salt-sensitive hypertensive rats. Several miRNAs influence *AGT* expression and are associated with cardiovascular diseases. Expression of both *ACE* and *ACE2* genes is regulated by DNA methylation, histone modifications, and miRNAs. Expression of both *angiotensinogen* and *CYP11B2* is reversibly regulated by epigenetic modifications and is related to salt-sensitive hypertension. The mineralocorticoid receptor (MR) exists in cardiovascular and renal tissues, in which many miRNAs influence expression and contribute to the pathogenesis of hypertension. Expression of the 11beta-hydroxysteroid dehydrogenase type 2 (*HSD11B2*) gene is also regulated by methylation and miRNAs. Epigenetic regulation of renal and vascular *HSD11B2* is an important pathogenetic mechanism for salt-sensitive hypertension.

## 1. Introduction

The renin–angiotensin–aldosterone system (RAAS) plays a pivotal role in the overall pathophysiology of hypertension [1,2,3]. Angiotensinogen (AGT) is the only known substrate of renin and is the rate-limiting enzyme of the RAAS. The levels of AGT are able to control the activity of the renin–angiotensin system; its upregulation may lead to elevated angiotensin II levels and blood pressure and has been implicated in cardiovascular and renal injuries [4]. There is growing evidence that adipose AGT has a potent role in the development of hypertension [5,6].

Angiotensin-converting enzyme (ACE) plays a major role in the RAAS (Figure 1). The central function of ACE is the conversion of angiotensin I to II. Tissue ACE is recognized as a key factor in cardiovascular and renal diseases. Pathological activation of local ACE has harmful effects on the cardiovascular tissues and kidneys [7,8]. By generating the vasodilator Ang-(1–7) and hydrolyzing portion of angiotensin II, ACE2 counterbalances the vasopressive effect of ACE [9]. ACE2 is the receptor for entry of SARS-CoV-2, which is the cause of COVID-19 in humans. It is expressed in cardiovascular and renal tissues and is related to the complications of COVID-19 infection [10,11].

Both angiotensin II type 1 receptor (AT1R) and mineralocorticoid receptor (MR) have well-founded functions in vasoconstriction, cellular proliferation, inflammation, and fibrosis. Treatment with AT1R blockers or MR antagonists (MRAs) protects against cardiovascular and renal injuries in patients with hypertension or diabetes mellitus [12,13,14]. Aldosterone is produced in the zona glomerulosa of the adrenal cortex by aldosterone synthase (CYP11B2) and is known to promote cardiac fibrosis and hypertrophy with concurrent elevation of inflammatory and oxidant signaling [15]. Patients with primary aldosteronism have a higher incidence of myocardial infarction and stroke than patients with essential hypertension [16,17]. Experimental animal data support the role of aldosterone in mediating cardiovascular and renal injury. In the salt-sensitive hypertensive (SSH) rat, administration of the mineralocorticoid receptor antagonist (MRA) greatly attenuated cardiac hypertrophy [18]. An important pathological effect of aldosterone in the heart has also been reported in experimental models of mineralocorticoid hypertension [19]. In these studies, prolonged exposure to aldosterone was associated with the development of myocardial hypertrophy and fibrosis. Local synthesis of aldosterone or angiotensin II has been reported [20,21]. Aldosterone produced in cardiovascular or renal tissues contributes to the development of or complications resulting from hypertension [22,23].

In large populations, significant correlations between salt intake, blood pressure, and hypertension incidence have been reported [24]. Salt-sensitive hypertension (SSH) is defined as a 10% increase in mean blood pressure due to a high-salt diet [25]. The proportion of salt-sensitive hypertension (SSH) is about 50% of hypertension and is associated with an increased risk of cardiovascular and renal injuries [26].

## 2. Epigenetic Regulation of Gene Expression

Epigenetic changes are inherited modifications that are not part of the DNA sequence. Gene expression is regulated at various levels, including via DNA modifications. Of these modifications, histone acetylation regulates gene expression [27], and DNA hypermethylation induces gene silencing [28]. Gene expression is also regulated by RNA modifications, which mediate RNA metabolism [29].

### 2.1. DNA Methylation

DNA methylation is generally involved in stabilizing the silent state of genes by either blocking DNA-binding transcription factors or recruiting methyl-CpG-binding domain (MBD) proteins, which favor the formation of transcriptionally inactive forms of chromatin (heterochromatin) [30]. Among the MBD proteins (methyl-CpG-binding proteins), MBD1 and MBD2 repress the transcription from methylated gene promoters. DNA methylation at the 5′-cytosine of CpG dinucleotides is a major epigenetic modification in eukaryotic genomes that is required for mammalian development [28], and it is associated with the formation of heterochromatin and gene silencing.

Dysregulated DNA methylation of renin–angiotensin system genes is involved in the pathogenesis of hypertension and cardiovascular diseases [30]. DNA methylation is established during normal development as well as disease progression. However, the DNA methylation pattern often changes dynamically in response to environmental changes [28,31]. Cardiovascular disorders, diabetes mellitus, and dyslipidemia, as well as lifestyle changes, affect DNA methylation dynamically.

### 2.2. Histone Modifications

Histone modifications are epigenetic modifications characterized by the addition of an acetyl group to histone proteins, specifically at lysine residues within the histone N-terminal tail [27]. Histone modifications are catalyzed by histone acetyl transferases (HATs) and histone deacetylases (HDACs), which are associated with transcription factors (TFs) [32]. Huang et al. [33] reported increased aldosterone production in rodents deficient in histone demethylase lysine-specific demethylase 1.

### 2.3. Micro RNAs (miRNAs)

miRNAs are small, non-coding RNA molecules, approximately 22 nucleotides in length, that regulate gene function at the post-transcriptional level [29]. These small RNAs act via complementary binding to the 3′-UTR and occasionally the 5′-UTR or coding regions of target miRNAs [29]. miRNAs are associated with several cardiovascular and renal diseases [34,35]. For example, miRNA-21 and miRNA-155 are associated with atherosclerosis, neovascularization, and vascular remodeling.

## 3. Epigenetic Regulation of the *AGT* Gene

The human *AGT* promoter possesses a number of CpG dinucleotides that are targets of DNA methylation. The human *AGT* promoter, which is located near a CCAAT enhancer-binding protein (CEBP)-binding site containing a CpG dinucleotide at positions −218/−217, is hypomethylated in the liver, heart, and HepG2 hepatocytes with high *AGT* expression [30]. In cultured human cells, interleukin 6 stimulation induced DNA demethylation near a CEBP-binding site and a transcription start site; this demethylation was accompanied by increased CEBP-β recruitment and chromatin accessibility of the *AGT* promoter. The methylation status of the CpG dinucleotide within the CEBP-binding site is inversely associated with *AGT* expression [30]. DNA demethylation causes a shift in the *AGT* expression phenotype from the inactive to the active state.

Several miRNAs influence *AGT* and cause cardiovascular diseases. Sharma et al. [36] reported that in heart failure, *AGT* expression was upregulated in the hypothalamus via a post-transcriptional mechanism mediated by miRNA-133a. Wang et al. [37] reported that miRNA-149-5p affected *AGT* expression, which promoted inflammatory responses. miRNA-133b induces proliferation and inhibits apoptosis in retinal endothelial cells by targeting *AGT* [38]. However, miRNA-29a inhibits retinal neovascularization to prevent the development and progression of retinopathy via downregulation of *AGT* [39]. miRNA-31/-584 binds to the *AGT* and influences coronary artery disease [40].

### 3.1. Salt-Sensitive Hypertension (SSH)

We showed that high salt intake reduces the levels of circulating RAAS but increases those of tissue RAAS in SSH rats [18,41]. Transgenic mice expressing human aldosterone synthase exhibit SSH [42]. Tissue AGT is an important effector molecule for blood pressure regulation.

Overexpression of *AGT* in the heart increases blood pressure and cardiac hypertrophy, and young spontaneously hypertensive rats show elevated tissue *AGT* expression [43]. High salt intake increases the cardiac mRNA levels of *AGT* as well as *AT1R* in SSH rats [18]. Treatment with MRAs decreases tissue AGT levels and improves cardiovascular injuries independent of blood pressure [18].

High salt intake demethylates the *AGT* promoter in the heart of SSH rats. In treatments with the MRA, eplerenone decreases the *AGT* mRNA level and methylates the *AGT* promoter in SSH rats [41]. DNA demethylation occurs around the transcription start site and CEBP-binding sites. These results suggest that a salt-associated stimulatory signal may recruit CEBP to its binding sites within the first exon to activate *AGT* transcription. This mechanism explains the beneficial effects of MRAs on cardiovascular diseases. Based on the role of epigenetics in the development of chronic cardiovascular and metabolic diseases, it is presumed that epigenetic intervention may be an effective strategy for the treatment of these diseases.

### 3.2. Primary Aldosteronism

More and more studies are being conducted on primary aldosteronism (PA), which accounts for 5–10% of the hypertensive population [44]. Increased prevalences of diabetes mellitus and metabolic syndrome in patients with PA were reported in a large cohort study [45,46]. A high level of aldosterone in individuals without insulin resistance at baseline was found to predict the development of insulin resistance 10 years later [47]. Experimental and clinical evidence indicates that a high aldosterone level impairs glucose metabolism by inhibiting insulin secretion and increasing insulin resistance [48]. Wu et al. [49] demonstrated increased inflammation and fibrosis in peripheral adipose tissues in patients with aldosterone-producing adenoma. Kalupahara et al. [50] reported that overproduction of *AGT* in adipose tissues induces adipose inflammation, glucose intolerance, and insulin resistance.

Excess circulating aldosterone upregulated *AGT* expression and was accompanied by DNA hypomethylation around a CEBP-binding site and a transcription start site in human visceral adipose tissue [30]. Increased *AGT* expression from visceral adipose tissue may contribute, in part, to the development of hypertension and metabolic abnormalities in PA [28].

## 4. Epigenetic Regulation of the Angiotensin-Converting Enzyme (ACE)

ACE plays a central role in the generation of angiotensin II and the degradation of bradykinin, thereby influencing blood pressure regulation and vascular remodeling [51]. Alterations in endothelial ACE expression or activity are associated with inflammatory cardiovascular diseases, including hypertension, diabetes, and atherosclerosis [52]. The human *ACE* promoter contains CpG islands [53], and hypomethylation of the *ACE* has been linked to fetal programming and the potential development of future diseases [54].

Somatic ACE is crucial in cardiovascular homeostasis and displays a tissue-specific profile [55]. We have reported an increased ACE mRNA level in the heart and kidney of SSH rats [18]. Epigenetic patterns modulate gene expression, the alterations of which have been implicated in pathologies, including hypertension. The effects of a maternal low-protein diet on the development of hypertension and cardiovascular diseases during adulthood have been documented extensively [56]. Goyal et al. [57]. reported that a maternal low-protein diet increased the levels of *ACE* mRNA and demethylated CpG islands of *ACE* promoter in the brain. Riviere et al. [58] reported that the methylation of the *ACE* promoter influenced mRNA levels in the rat lung and liver but not the kidney. They concluded that the basal methylation pattern of the *ACE* promoter correlates with somatic *ACE* transcription.

The expression of *ACE* in tissues is also controlled by histone modifications and miRNAs. Lee et al. [59] reported that *ACE* is upregulated in the heart and kidney of spontaneously hypertensive rats (SHRs) via histone code modifications. Hu et al. [60] reported that mechanical stretch suppresses miRNA-145 expression and promotes *ACE* expression to alter the vascular smooth muscle cell phenotype. miRNAs act as critical regulators of major cellular functions and are considered in the pathogenesis of hypertension. Kohlstedt et al. [61] reported that upregulation of miRNA-143/145 suppressed endothelial *ACE* expression. Post-transcriptional regulation of miRNAs in the blood vessels may contribute to the pathogenesis of hypertension.

## 5. Epigenetic Regulation of ACE2

Angiotensin II is an important vasoconstrictor, whereas angiotensin-converting enzyme 2 (ACE2) promotes vasodilation by degrading angiotensin II and generating the vasodilator Ang 1–7 [62]. Increased expression of *ACE2* protects against elevated blood pressure, whereas *ACE2* inhibition or deletion promotes hypertension [63].

DNA methylation is an important mechanism of *ACE2* regulation. Goyal et al. [64] showed hypomethylation together with high expression of *ACE2* in lung epithelial cells. We found that the *ACE2* mRNA level was significantly decreased in the hearts of SSH rats compared with control rats [18]. However, the methylation ratio did not differ between SSH rats and control rats (Figure 2) [65].

Histone modifications within the *ACE2* gene region have been reported in COVID-19 infection [66,67,68]. Pinto et al. [69] reported an elevated *ACE2* mRNA level with histone modifications such as histone acetyltransferase 1 and adenosine deaminase 2 in the lungs of patients with severe COVID-19 infection. There are no reports of histone modifications of *ACE2* in cardiovascular diseases, including hypertension.

There have been several reports on the roles of miRNAs in the regulation of *ACE2* [70,71,72]. Gu et al. [73] reported that exercise training decreased blood pressure and increased *ACE2* and miRNA-143 expression levels in the aorta in SHRs. Wang et al. [74] reported that angiotensin-(1–7) decreased vascular inflammation and improved vascular function by modulating the expression of miRNA-146a in human aortic endothelial cells. Treatment with an angiotensin II receptor blocker was reported to modulate the level of miRNA-146a/b, along with improvement of the *ACE2* level and attenuation of vascular remodeling in hypertension [75]. We have reported that treatment with MRAs improved cardiac hypertrophy and increased *ACE2* mRNA in the hearts of SSH rats [18]. These data suggest that RAAS blockers exert cardiovascular protective effects via ACE2 signaling and miRNA levels.

## 6. Epigenetic Regulation of AT1R

Angiotensin II increases blood pressure as well as cardiovascular and renal tissue injuries via AT1R. *AT1R* expression is regulated by DNA methylation and miRNAs [76,77]. Kawakami-Mori et al. [78] reported that in the offspring of pregnant rats receiving a low-protein diet or dexamethasone, the mRNA level of the AT1R gene (*Agtr1a*) was increased in the hypothalamus, concurrent with hypomethylation of the *Agtr1a* promoter. These offspring showed SSH. Ghosh et al. [79] found that *Agtr1a* expression in the hypothalamus progressively increased, while the methylation status of the *Agtr1a* promoter decreased in SHRs compared with Wistar–Kyoto rats. Thus, epigenetic modulation of hypothalamic *Agtr1a* contributes to SSH or essential hypertension.

Exercise is one of the most effective treatments for hypertension. Shan et al. [80] reported that maternal exercise upregulates the DNA methylation of *Agtr1a* and decreases gene expression in mesenteric arteries in offspring SHRs. Maternal exercise reduces blood pressure and cardiovascular reactivity of the offspring from SHRs.

Several miRNAs negatively regulate *Agtr1a* expression at the post-transcriptional level. Zheng et al. [81] demonstrated that miRNA-155 suppressed the activity of an *Agtr1a* 3′-UTR reporter construct via a luciferase assay. They also reported that miRNA-155 plays an important role in regulating adventitial fibroblast differentiation and contributes to the suppression of *Agtr1a* expression. Cross-talk between aldosterone and angiotensin II has been proposed in the pathogenesis of cardiovascular and renal diseases [82,83]. DuPont et al. [84] reported that the vascular MR regulates miRNA-155 and *Agtr1a* to promote vasoconstriction and elevate blood pressure.

AT1R-associated protein is a direct binding protein of AT1R that acts as an endogenous inhibitor of hypertension pathogenesis in cardiovascular and renal tissues [85,86,87,88]. Hirota et al. [88] reported that miRNA-125a-5b/125b-5b contributes to the pathological activation of AT1R-associated protein in mouse distal convoluted tubule cells.

## 7. Epigenetic Regulation of *CYP11B2*

*CYP11B2* expression is epigenetically regulated by DNA methylation and miRNAs. Angiotensin II increases *CYP11B2* expression and aldosterone synthesis both in the adrenal gland and cardiovascular and renal tissues. We have reported that angiotensin II infusion in rats induced hypomethylation of the *CYP11B2* promoter and increased gene expression in the adrenal gland [89]. A low-salt diet decreases the methylation ratio of rat *CYP11B2* and increases the *CYP11B2* mRNA level in parallel with aldosterone synthesis. Interestingly, switching from a low-salt diet to a high-salt diet resulted in a change from a *CYP11B2* hypomethylated to a hypermethylated state [89]. These results suggest that angiotensin II regulates aldosterone synthesis by the mechanism of DNA methylation.

Both miRNA-10b and miRNA-24 are negative regulators of the cortisol synthase genes *CYP11B1* and *CYP11B2* [90,91]. Zhang et al. [92] reported that miRNA-193a-3p not only downregulated *CYP11B2* expression but also acted as a tumor suppressor in aldosterone-producing adenoma. miRNA-124a-5p and miRNA-124b-5p are also negative regulators of *CYP11B2* [91]. Syed et al. [93] reported that excess aldosterone increased miRNA-21 expression in the rat heart.

### 7.1. Epigenetics and Aldosterone-Producing Adenoma (APA)

The most common clinical subtypes of PA are APA and bilateral adrenocortical hyperplasia [94]. We and others have reported a lower level of *CYP11B2* methylation in APAs than in adrenal tissues or non-functioning adrenal adenomas. A negative correlation between the *CYP11B2* methylation ratio and mRNA level was found [95,96,97]. Epigenetic control of *CYP11B2* expression may play an important role in aldosterone synthesis in APAs. We found a *KCNJ5* mutation in aldosterone-producing microadenoma and aldosterone-producing cell clusters, in which the methylation rate of *CYP11B2* was decreased compared with adjacent adrenal tissues [98]. Further study is necessary to clarify the mechanism of aldosterone overproduction, including the epigenome and metabolome, in aldosterone-producing cell clusters and APA.

### 7.2. Epigenetic Regulation of Mineralocorticoid-Related Genes in SSH

Mineralocorticoids, including aldosterone, are an important pathological factor in SSH [99]. Epigenetic mechanisms involved in the development of SSH have been reported [100]. Maternal lipopolysaccharide exposure during pregnancy induces upregulation of *Rac1* via histone modifications mediated by H3K9me2 across generations, resulting in salt-induced activation of the Rac1-/MR pathway in the kidney and development of SSH [101]. We reported that local RAAS activation caused SSH. A high-salt diet-induced hypomethylation of *CYP11B2* in the hearts of SSH rats and increased cardiac aldosterone synthesis and hypertrophy [102]. MRA treatment not only decreased blood pressure but also induced hypermethylation of *CYP11B2* in the heart [102].

Aldosterone synthesis and *CYP11B2* expression are upregulated in cardiac tissues during hypertrophic cardiomyopathy (HCM) and are recognized as major HCM phenotype modifiers [102]. Aldosterone directly affects cardiac hypertrophy and fibrosis. We previously reported that aldosterone, locally produced in cardiovascular tissues, exerts its effects via paracrine or intracrine mechanisms [103]. Garnier et al. [104] reported coronary endothelium-independent dysfunction without hypertrophy in the hearts of transgenic mice overexpressing *CYP11B2*. Alesutan et al. [105] showed *CYP11B2* expression in human coronary arteries as well as smooth muscle cells. In their study, the *CYP11B2* mRNA level was higher in aortic tissues in klotho-hypomorphic mice than in control mice, and spironolactone ameliorated aortic osteoinductive reprogramming in adrenalectomized klotho-hypomorphic mice. We found that treatment with spironolactone improved cardiac hypertrophy in adrenalectomized hypertensive rats [106]. Yoshimura et al. [107] reported increased *CYP11B2* expression in the hearts of patients with cardiac failure, while we found a clear association between CpG methylation and *CYP11B2* expression in the cardiac tissues of HCM patients [102]. Hypomethylation of the *CYP11B2* promoter aberrantly increases *CYP11B2* expression, which induces cardiac hypertrophy or cardiomyopathy. The molecular mechanisms regulating demethylation of *CYP11B2* in the heart remain unclear.

### 7.3. Epigenetic Control of Mineralocorticoid Receptors

Mineralocorticoid receptors (MRs) exist in both epithelial and non-epithelial cells. MRs in vascular endothelial cells and smooth muscle cells (VSMCs) are involved in vascular smooth muscle hypertrophy and endothelial dysfunction [108]. Cardiac MR contributes to cardiac tissue inflammation, fibrosis, and cardiac dysfunction [109]. We detected MR mRNA in VSMCs, and aldosterone increased the incorporation of tritiated leucine into these VSMCs; this incorporation was inhibited by a specific aldosterone antagonist [110]. MR activation in VSMCs or endothelial cells increased oxidative stress mediated by activation of NADPH oxidases [108]. Oxidative stress promotes the proliferation of VSMCs and regulates blood pressure. Mesquita et al. [111] reported that MR signaling activates long cardiac Ca1,2 N-terminal mRNA expression via P1-promoter activation, leading to hypertension. MR activation in VSMCs also induces the expression of collagens 1 and 3, IL-16, CTLA4, and genes associated with vascular calcification [112].

MRs are epigenetically controlled by methylation, histone modifications, and miRNAs. The histone deacetylase 3/4 complex stimulates the transcriptional activity of MRs [113]. A maternal high-fat diet upregulates MR function in the offspring’s blood vessels via an epigenetic mechanism [114]. Camarda et al. [115] reported that MR-knockout mice or blockade increased the levels of the enhancer of zeste homolog 2 and histone-H3 lysine-27-specific methyltransferase and prevented vascular stiffness and fibrosis.

microRNAs have been implicated in multiple MR-related cardiovascular and renal injuries. Soeber et al. [116] reported that miRNA-124 and miRNA-135a are potential regulators of MR gene expression. miRNA-21 expression is upregulated in the heart by excess aldosterone. Genetic ablation of miRNA-21 exacerbates cardiac hypertrophy and injury in mineralocorticoid-excess mice [93]. miRNA-31 targets the 3′-UTR of MR as well as cardiac troponin-T. Inhibition of miRNA-31 improves cardiac dysfunction and prevents the development of post-ischemic adverse remodeling [117]. Garg et al. [118] found that miRNA-181a is a novel regulator of MR-mediated cardiac remodeling. MR activation increases miRNA-204 expression, which induces T-type calcium channel expression in cardiomyocytes [119]. DuPont et al. [84] reported that the age-associated decrease in miRNA-155 in mesenteric arteries was associated with increased expression of MRs and L-type calcium channels, which cause hypertension. miRNA-34 has been reported to be dysregulated in various human cancers and to play a tumor-suppressive role because of its synergistic effect with the well-known tumor suppressor p53. The role of miRNA-34b/C in MR-mediated VSMC calcification has been reported. Treatment with the MR antagonist upregulates miRNA-34b/c and inhibits vascular calcification [120]. miRNA-766 targets MR genes directly and induces an anti-inflammatory effect via inhibition of NF-kB signaling [121]. NF-kB binds to MR promoters and decreases MR isoform levels.

MRs contribute to hypertension by increasing renal salt reabsorption and promoting kidney dysfunction via direct effects on renal parenchymal cells. We have reported that the MR antagonist, eplerenone, prevented renal injury and decreased blood pressure in salt-sensitive hypertensive rats [122]. The miRNA-466 family targets and regulates the expression of MRs and serum glucocorticoid-regulated kinase 1 (SGK1) [123], which stimulates MR-dependent renal sodium reabsorption, increases blood pressure, and promotes kidney inflammation and fibrosis [124]. Park et al. [125] reported that in aldosterone-treated cells, reduced miRNA-34c-5p level increased Ca^2+^/calmodulin-dependent protein kinase type II beta-chain expression and stimulated fibronectin and alpha-smooth muscle actin, which play a significant role in the development of fibrosis.

Long non-coding RNAs (lncRNAs) interact with proteins and interfere with miRNAs by acting as molecular sponges to modify the epigenome. Zhang et al. [126] reported that lncRNA Tug1 promotes angiotensin II-induced renal fibrosis by binding MRs and negatively regulating miRNA-29b-3p. Upregulation of lncRNA H19 is reported to contribute to aldosterone-MR complex-promoted vascular smooth muscle calcification by sponging miRNA-106a-5p [127].

Aldosterone inhibits miRNA-192, which increases the serine/threonine kinase with no lysine (WNK1) [128]. Long-form WNK1 is an important regulator of both K^+^ and Na^+^ transport [129]. Both miRNA-194 and miRNA-802 regulate renal outer medullary potassium channels [130,131]. Edinger et al. [132] reported that aldosterone downregulated mmu-miRNA-335-3p, mmu-miRNA-290-5p, and mmu-miRNA-1983, which increased epithelial sodium channel-mediated sodium transport in mouse cortical collecting ducts.

### 7.4. Epigenetic Control of 11ß-Hydroxysteroid Dehydrogenase Type 2

The enzyme 11ß-hydroxysteroid dehydrogenase (11ß-HSD) catalyzes the conversion of glucocorticoids into their inactive metabolites and modulates mineralocorticoid and glucocorticoid activity. (Figure 3) Biochemical studies have revealed the existence of two isoforms of 11ß-HSD, NAD^+^-dependent and NADP^+^-dependent. 11ß-HSD2 (the NAD^+^-dependent isoform) is found in distal portions of the nephron, which co-localizes with MRs. Kidney-specific gene deletion of *HSD11B2*, which induces human or renal dysfunction, causes hypertension [133,134]. We have reported that vascular 11ß-HSD2 contributes to salt-sensitive hypertension [135].

*HSD11B2* is epigenetically controlled by methylation and miRNAs. Alikhani-Koopaei et al. [136] reported that methylation of the promoter region of *HSD11B2* regulates *HSD11B2* gene expression. Nuclear factor 1 (NF1) is a strong stimulator of the *HSD11B2* gene, and this effect is dependent on the position and the combination of methylated CpGs. Apparent mineralocorticoid excess is a rare genetic hypokalemic low-renin hypertension. Mutation of the *HSD11B2* gene was reported [137]. Pizzolo et al. [138] reported that the hypertension phenotype of apparent mineralocorticoid excess was associated with higher methylation of the *HSD11B2* promoter region compared with normotensive heterozygous relatives.

*HSD11B2* expression is regulated by several miRNAs. Rezaei et al. [139] found lower expression of mo-miRNA-20a-5p, mo-miRNA-19b-3p, and mo-miRNA-190a-5p in Sprague Dawley rats compared with Wistar rats, and uninephrectomy decreased the expression of mo-miRNA 26b-5p, mo-miRNA-19b-3p, and mo-miRNA-29b-3p in Sprague Dawley rats. They also showed reduced 11β-HSD2 activity after miRNA-20a overexpression. High fructose consumption is related to hypertension and obesity [140]. Nouchi et al. [141] reported that although maternal high-fructose corn syrup did not affect the methylation status of *HSD11B2*, it increased miRNA-27a-5p overexpression and decreased mRNA expression in the kidney of the offspring.

## 8. Conclusions

The epigenetic modifications of local RAAS in cardiovascular, renal, and adipose tissues and their influence on hypertension are described. Gene expression of RAAS is regulated by epigenetic modifications such as DNA methylation, histone modifications, and miRNAs. In salt-sensitive hypertension, hypomethylation of *AGT* and *CYP11B2* increases both mRNA levels in cardiovascular tissues. miRNAs regulate the gene expression of MR and *HSD11B2* in the kidney, which controls blood pressure and electrolytes. Epigenesis of RAAS needs to be further clarified both under normal physiological conditions and in pathophysiological states, including hypertension.

## Figures and Tables

**Figure 1 ijms-25-08099-f001:**
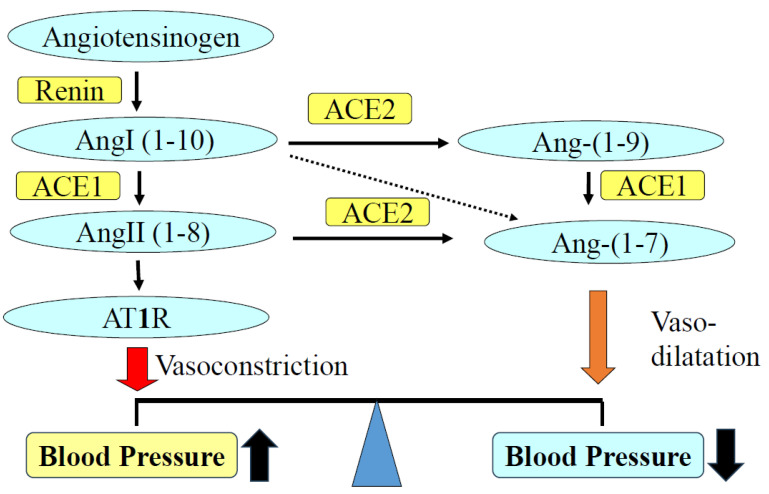
Role of the ACE and ACE2/Ang-(1–7) in hypertension. ACE converts angiotensin I to angiotensin II. Angiotensin II increases blood pressure as well as injuries to cardiovascular and renal tissues via AT1R. ACE promotes vasodilation by degrading angiotensin II and generating vasodilator Ang 1–7. ACE, angiotensin-converting enzyme; Ang, angiotensin; AT1R, angiotensin II type 1 receptor.

**Figure 2 ijms-25-08099-f002:**
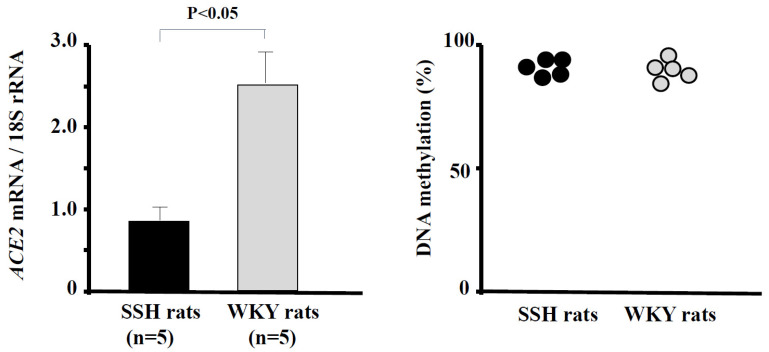
*ACE2* mRNA levels and methylation ratios in the hearts of SSH rats and control rats. The *ACE2* mRNA level was significantly lower in SSH rats than in control rats (*p* < 0.05). The methylation ratio of CpGs of promoter of *ACE2* in the hearts did not differ between SSH rats and control rats. (Data were cited from [65]). ACE, angiotensin-converting enzyme; SSH, salt-sensitive hypertension.

**Figure 3 ijms-25-08099-f003:**
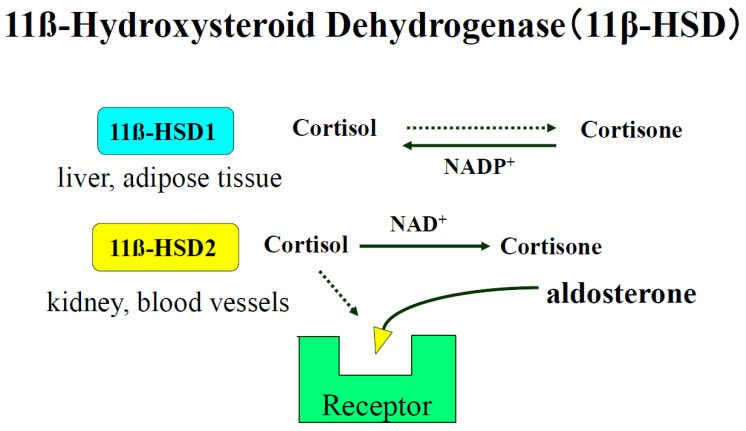
Biochemical studies have revealed that there are two isoforms of 11ß-HSD, a NAD^+^-dependent form (11ß-HSD2) and a NADP^+^-dependent form (11ß-HSD1). 11ß-HSD2 is found in tissues with high levels of MR activity, such as kidney, placenta, and colon [59]. 11ß-HSD2 is also found in blood vessel type 1 11β-HSD (11β-HSD1), which is highly expressed in adipose tissue, liver, and skeletal muscle and plays a central role in obesity, diabetes mellitus, and hypertension.

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
