# Peer review of "Epigenetic Regulation of the Renin–Angiotensin–Aldosterone System in Hypertension"

_ijms, 2024, doi:10.3390/ijms25158099_

Round 1

Reviewer 1 Report

Comments and Suggestions for Authors

This is an interesting review covering a fontier and growing field: that of epigenetic regulation of biological and physiological conditions. 

I strongly recommend the inclusion of 3 more figures related with the crosstalk between epigenetic regulation and renin-aldosterone system associate signaling pathways in cardiovascular, renal and adipose tissue. Visceral adipose tissue deserves special atention. In the case of renal tissue I suggest special attention to the distal tubule and sodium trnasporters

Comments on the Quality of English Language

Minor revision is needed with a native English speaker.

Author Response

Thank you for your comments. I added figure 3. I added the content of the distal tubule and sodium transport in the text.

Reviewer 2 Report

Comments and Suggestions for Authors

Takeda and his co-authors conducted a comprehensive review on the epigenetic changes associated with renin-angiotensin-aldosterone regulaiton in hypertension. They examined how mechanisms such as DNA methylation, histone post-translational modifications, and microRNA regulation contribute to hypertension in both animals and humans, demonstrating that these changes can be heritable yet modifiable. The review identified specific target organs and tissues undergoing epigenetic changes and discussed conditions during pregnancy that increase susceptibility to hypertension in offspring. They also explored behaviors like salt intake and physical activity that can influence blood pressure through epigenetic modifications. Of clinical importance, the authors found that mineralocorticoid receptor antagonists and angiotensin receptor blockers can beneficially affect the pathophysiological processes of the disease.

Figure 1 effectively illustrates the rate-limiting role of angiotensinogen in hypertension and the roles of ACE and ACE2 in vasoconstriction and vasodilation. 

Despite the review's valuable insights for both preclinical research and clinical practice, there are several major and minor concerns regarding the article.

Major Concerns:

  1. Structure:
    • The review lacks a clearly defined research question, and the methodology is completely omitted. It is essential to describe the databases searched, the search strategy, and the search terms used. Additionally, the quality assessment of the included articles should be explained. The authors cited their previous work and included some results (Figure 2) from these studies. They should detail how their perspectives and experiences informed their decisions, including the sampling strategy. 

To address these issues, it is recommended to follow guidelines on writing a narrative review:https://link.springer.com/article/10.1007/s00296-011-1999-3 .

    • Some sections could be expanded with the authors' opinions and hypotheses for future research. The conclusions could also be more detailed.
  1. Abstract:
    • The abstract should include the research question, methods, and conclusions.
  2. Figure 3:
    • Figure 3 is missing; only the explanation is provided.

Minor Concerns:

  1. Introduction:
    • The introduction should emphasize the importance of both circulating and local aldosterone in the pathophysiology of cardiovascular disorders (lines 52-54).
    • Paragraph concerning introduction on salt sensitive hypertension lines 109-118 in my opinion better suites in the introductionn.
    • The section on ACE2 as an SRS-CoV-2 receptor, while interesting, is not within the scope of the research topic (lines 39-41, 192-194).
  2. Typographical Errors:
    • Line 89: Typo in the reference, "429" should be "29."
    • Line 37: Typo, " [7. 8]" should be " [7, 8]."
    • Line 150: Typo, "emzyme" should be "enzyme."
    • Line 159: Typo, "Epigenetic patterns modulate gene expression. alterations of which have been implicated in pathologies including hypertension." should be "Epigenetic patterns modulate gene expression, alterations of which have been implicated in pathologies including hypertension." 
    • Line 250: Add the abbreviation APA or explain it in line 251.
    • Line 259: Typo, "thatinvolving" should be "that involving."
    • Line 267: Typo, "S high salt diet" should be "a high salt diet."
    • Line 374: Typo, "miTNA" should be corrected (please clarify the correct term if it is an error). 
  3. Reference Formatting:
    • Reference 97: Add the complete DOI: "doi: 10.1016/j.steroids.2019.108470."
    • Formatting issues in references 34, 39, 41, 42, 44-56, 59-64, 66-70, 73, 75-77, 79-80, 82, 84-86. (some parts are underlined)
  4. Content Clarification:
    • Lines 93-94: Clarify if it is necessary to name both liver and HepG2 hepatocytes.
    • Paragraph on salt-sensitive hypertension (lines 109-118) might be better suited in the introduction.
    • Lines 121 and 124: Address redundancy. "High-salt intake increases the cardiac mRNA levels of AGT as well as AT1R in SSH rats [38]" and "High-salt intake increases the AGT mRNA level in the heart and demethylates the AGT promoter in SSH rats."
    • Consider discussing the disrupter of telomeric silencing (Dot1) with intrinsic histone methyltransferase (HMT) activity, which interacts with the Af9 gene, producing high sodium channel permeability and silencing the hydroxysteroid dehydrogenase-11β2 gene, thereby preventing cortisol metabolism to cortisone and overstimulating aldosterone receptors (refer to DOI: 10.1007/s11906-010-0173-8). Is this the same channel malfunction as in CKNJ5 mentioned in lines 256-258?

By addressing these major and minor concerns, the review will be more comprehensive and provide clearer guidance for both researchers and clinicians.

Author Response

Thank you very much for your comments. I modified the abstract, however, my review is not systematic review. I added the explanation in Figure 2 legend. I added several sentences of aldosterone in the pathophysiology in "Introduction". I moved the text of salt sensitive hypertension to "Introduction".   Methylation of ACE2 gene in hypertension should be clarified as your suggestion. I modified several typoerrors.  

Round 2

Reviewer 2 Report

Comments and Suggestions for Authors

Dear authors,

Please response to my extensive review in point-by-point manner addressing in details all of the points. 

Author Response

Thank you for your comments. My review is narrative, not systematic. In Figure 2 legend I added the source of the figure. I added Figure 3. I modified the abstract. I added the sentences of the importance of aldosterone in "Introduction" and also moved the sentence of "salt-sensitive hypertensin " to "Introduction".   Although there is no methylation data of ACE2 gene in hypertension, it may become to be important. I corrected several typoerrors 
